# Worker ants promote outbreeding by transporting young queens to alien nests

Mathilde Vidal [1✉], Florian Königseder[1], Julia Giehr[1], Alexandra Schrempf[1], Christophe Lucas [2] & Jürgen Heinze [1]

Choosing the right mating partner is one of the most critical decisions in the life of a sexually reproducing organism and is the basis of sexual selection. This choice is usually assumed to be made by one or both of the sexual partners. Here, we describe a system in which a third party – the siblings – promote outbreeding by their sisters: workers of the tiny ant *Cardiocondyla elegans* carry female sexuals from their natal nest over several meters and drop them in the nest of another, unrelated colony to promote outbreeding with wingless, stationary males. Workers appear to choose particular recipient colonies into which they transfer numerous female sexuals. Assisted outbreeding and indirect female choice in the ant *C. elegans* are comparable to human matchmaking and suggest a hitherto unknown aspect of natural history – third party sexual selection. Our study highlights that research at the intersection between social evolution and reproductive biology might reveal surprising facets of animal behavior.

[1] Chair of Zoology and Evolutionary Biology – University of Regensburg, Regensburg, Germany. [2] Institut de Recherche sur la Biologie de l'Insecte (UMR7261), CNRS – Université de Tours, Tours, France. ✉email: mathilde.vidal@biologie.uni-regensburg.de

Choosing the right mating partner is one of the most critical decisions in the life of a sexually reproducing organism. Mate choice is not only driven by the quality of potential mates but also affected by the genetic compatibility of mating partners, e.g.[1,2]. Mating with a too closely related individual typically results in inbreeding depression[3,4] and numerous organisms have evolved mechanisms to prevent sib-mating[5]. In ants and many other Hymenoptera inbreeding is particularly detrimental[6,7]. Due to their standard mechanism of sex determination, 50% of a female's eggs that are fertilized by sperm from a brother may develop into non-viable or sterile diploid males instead of diploid females[8,9], but see[10]. Sib-mating in social insects is therefore usually avoided, e.g., through an earlier maturation and farther dispersal of males, large nuptial flights, or individual colonies focusing on the exclusive production of one sex[11–13].

The ant genus Cardiocondyla is characterized by wingless "ergatoid" males, which mate with related female sexuals inside their natal nests[13–15]. Controlled breeding experiments showed that Cardiocondyla has evolved an alternative sex determination mechanism, in which neither brother–sister nor mother–son mating leads to the production of diploid males[16,17]. Nevertheless, prolonged inbreeding over ten generations in the laboratory led to shortened queen life span, higher brood mortality, and more strongly male-biased sex ratios[16]. This indicates that the natural life history of Cardiocondyla requires occasional outbreeding.

In most tropical Cardiocondyla, colonies may contain multiple fertile queens (polygyny). In these species, outbreeding appears to be facilitated by the occasional adoption of alien queens into established nests and the occurrence of winged disperser males[18–20]. In contrast, many Palearctic species of Cardiocondyla have evolved obligate monogyny, i.e., their colonies contain only a single fertile queen, and additional fertile queens are not tolerated[15,21,22]. Furthermore, the winged male phenotype has been lost completely in Palearctic Cardiocondyla[15,23]. Genetic data suggest that more than 2/3 of all matings in Palearctic Cardiocondyla involve close relatives[13,24,25]. This is in accordance with male winglessness and the observation that female sexuals, even though winged, do not engage in dispersal flights. However, the genetic data also raise the question of how 30% outbreeding is achieved.

The anecdotal observation of workers carrying winged female sexuals (gynes) between nests of C. elegans[24] suggested an exciting alternative way of inbreeding avoidance with a critical role of workers, which we here explore in detail by behavioral observations in the field and microsatellite genotyping. We hypothesize that gyne carrying serves to promote outbreeding and that workers as "royal matchmakers" transfer female sexuals into particular "target colonies." Our data show that most workers carry related gynes from their joint natal nest over several meters and drop them into the nest entrances of other colonies, where the gynes can mate with unrelated males. The occasional occurrence of unrelated pairs of workers and gynes indicates that there might be additional rounds of transport to further unrelated colonies. Gynes spend the winter in the nests of unrelated colonies and in spring disperse on foot to found their own colonies.

## Results

**Field observations.** In total, we mapped the location of 175 colonies in our collecting sites in Southern France (Table 1). Additional colonies were located, observed, and later excavated, but their position relative to other colonies was not exactly recorded. Raw data about field colonies can be found in the supplementary material[26].

**Table 1 Colonies and pairs of carriers and gynes sampled in a population of the ant Cardiocondyla elegans.**

**September 2014**

| Population | P | BN | CP | RFRK | SM |
|---|---|---|---|---|---|
| Colonies mapped | 9 | 8 | 8 | 10 | 5 |
| Pairs of GC observed | 0 | 0 | 1 | 0 | 1 |
| Pairs of GC observed with known destination | 0 | 0 | 0 | 0 | 0 |

**August 2015**

| Population | P | H | FK | BN | CP | RFRK | SM |
|---|---|---|---|---|---|---|---|
| Colonies mapped | 18 | 11 | 15 | 21 | 12 | 25 | 7 |
| Pairs of GC observed | 16 | 4 | 0 | 24 | 16 | 44 | 17 |
| Pairs of GC observed with known destination | 16 | 2 | 0 | 19 | 13 | 28 | 5 |

| Year-Month | April 2016 | July 2016 | August 2017 | August 2018 | August 2019 |
|---|---|---|---|---|---|
| Population | BN | BN | BN | RFRK | RFRK |
| Colonies mapped | 0 | 0 | 0 | 20 | 6 |
| Pairs of GC observed | 0 | 0 | 20 | 82 | 167 |
| Pairs of GC observed with known destination | 0 | 0 | 0 | 55 | 3 |
| destination | CP | CP | CP | RFRK | RFRK |

Colonies of the ant Cardiocondyla elegans mapped in the collection sites in France and the number of pairs of pairs of gyne (G) and carriers (C) that were observed in the field. In 2014 and 2015 several collecting sites were studied, which later were not investigated again. Due to changes in environmental conditions and to the short time range for collection, we focused in priority on populations BN, CP, SM, and RFRK.

**Table 2 Composition of colonies of the ant _Cardiocondyla elegans_.**

| Field collections | Number of colonies | Workers | Gynes | Males |
|---|---|---|---|---|
| September 2014 | 23 | 33 ± 33 (2–139) | 42 ± 56 (0–253) | 1.0 ± 1.4 (0–5) |
| August 2015 | 26 | 103 ± 92 (2–318) | 160 ± 125 (0–413) | 3.7 ± 4.4 (0–19) |
| April 2016 | 59 | 56 ± 42 (3–160) | 25 ± 28 (0–90) | 0 |
| July 2016 | 12 | 149 ± 75 (20–236) | 2 ± 3 (0–8) | 0 ± 1 (0–2) |
| August 2018 | 15 | 39 ± 22 (15–96) | 23 ± 31 (0–123) | 1.9 ± 3.1 (0–10) |
| August 2019 | 10 | 45 ± 18 (17–65) | 26 ± 21 (0–65) | 2.5 ± 2.9 (0–8) |

Number of colonies of the ant _Cardiocondyla elegans_ excavated, number of workers, gynes, and males found in the nest; mean ± SD (range) in different collection periods.

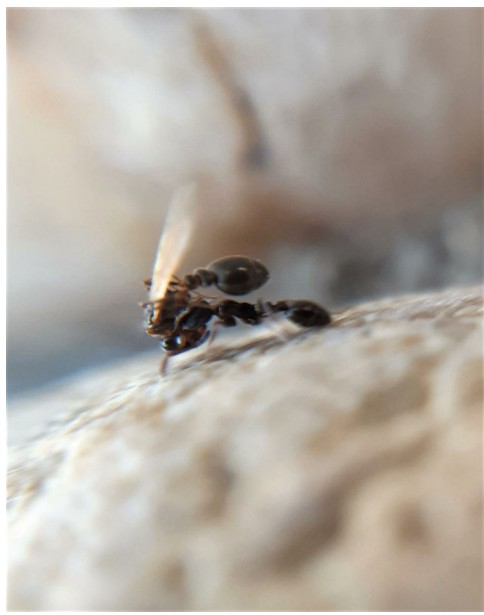

**Fig. 1 Gyne carrying in the ant _Cardiocondyla elegans_.** The inserts and the main picture show workers of the ant _Cardiocondyla elegans_ carrying winged female sexuals (gynes) to the nest entrance of another colony to allow outbreeding (photo by M. Vidal).

Nest densities locally reached five nests per square meter. Excavation of 145 colonies at the end of the observation periods (Table 2) revealed the presence of large numbers of gynes and males in a single chamber close to the nest entrance, just a few centimeters below the surface. Further vertical digging down to a depth of 1–2 m and in a diameter of about 40 cm around the nest entrance revealed many interconnected nest chambers, which contained mostly workers, brood, and the resident queen, but only few sexuals. In summer, workers were observed foraging for food solitarily between 8 a.m. and 1 p.m. and again between 4 p.m. and 9 p.m.

**Gyne carrying**. Workers were regularly observed carrying winged gynes over distances of several meters. Gyne carrying resembled the transport of adult ants during nest moving in other genera of the subfamily Myrmicinae[27]. A worker would drag a gyne out of the nest of the source colony with its mandibles, seize it by the head, and carry it over its back in a piggy-back like posture (Fig. 1).

In total, we followed the transport of 453 gynes, and for 141 pairs of carriers and carried gynes we could determine where transport had started and where it ended (Table 1; for details see supplementary material[26]). Gyne carrying was observed almost exclusively between 8 and 10 a.m. and from 12 to 2 p.m.

at air temperatures between 25 and 35 °C. A few worker–gyne pairs were observed until 3 p.m., in particular in the shade of trees, where ground temperatures were lower (~ 30 °C) than in sun-exposed areas (~50 °C). Gyne carrying was observed only twice in September 2014. In 2015, most gyne carrying was observed in July and August, but in July 2017, 2018, and 2019, carrying behavior appeared to be completely shifted to August because of hot temperatures. In July and August 2016, no gyne carrying was observed, perhaps due to relatively cool weather in July.

Workers carried gynes over several minutes and up to a maximum distance of 14.8 meters ($n = 182$, mean = 3.1 ± SD 0.2 m; range = 0.3–14.8 m) to a recipient colony, where they would dump it into the nest entrance. Workers apparently did not drop gynes randomly, e.g., into the nearest nest entrance, but instead passed several nests on their way to the selected recipient colony (mean = 2.3 ± SD 0.3 nests, $n = 57$ pairs in five collecting sites; Fig. 2). Furthermore, while _Cardiocondyla_ foragers typically search for food on tortuous paths[18], workers carried gynes in a more or less straight line to the recipient colony, making deviations only when forced by environmental obstacles, such as stones or dense tufts of grass.

Workers barely entered the nest of the recipient colony and always returned to the source colony. We could repeatedly observe that shortly thereafter another gyne was transported to the same recipient colony, probably by the same carrier (c.f. supplementary video 1[28]). In a few cases, two or three workers were observed carrying gynes on the same route or from the same source colony to different recipient colonies. Occasionally, the same recipient colony received gynes from different source colonies. The experimental removal of a carrier (e.g., for genetic analyses), showed that carriers were not immediately replaced by other workers. This suggested that in each colony only a small number of workers could be engaged in gyne carrying at a given time. In few cases, carriers were observed accidentally dropping the gyne. These rare events might explain the occasional observations of gynes walking solitarily outside the nests. Dissection of the genital apparatus of eight carried and four solitarily moving gynes in August 2015 revealed the presence of sperm in six carried and all four solitary queens.

Male carrying was observed only five times in September 2014 and twice in August 2015, but only for one male it could be determined that it was transported from one colony to another over a 3 m distance. In addition, we observed twice carrying of a worker and five times carrying of pupae or prepupae.

**Population structure and carrier–gyne relatedness**. The mean relatedness of nestmate workers was significantly lower than the value of 0.75 expected for full sisters, matching the assumption of monogyny and polyandry (Table 3; for the raw data see supplementary material[26]). Following equation 5 from Tarpy and Nielsen (2002)[29] the relatedness estimates translate into effective

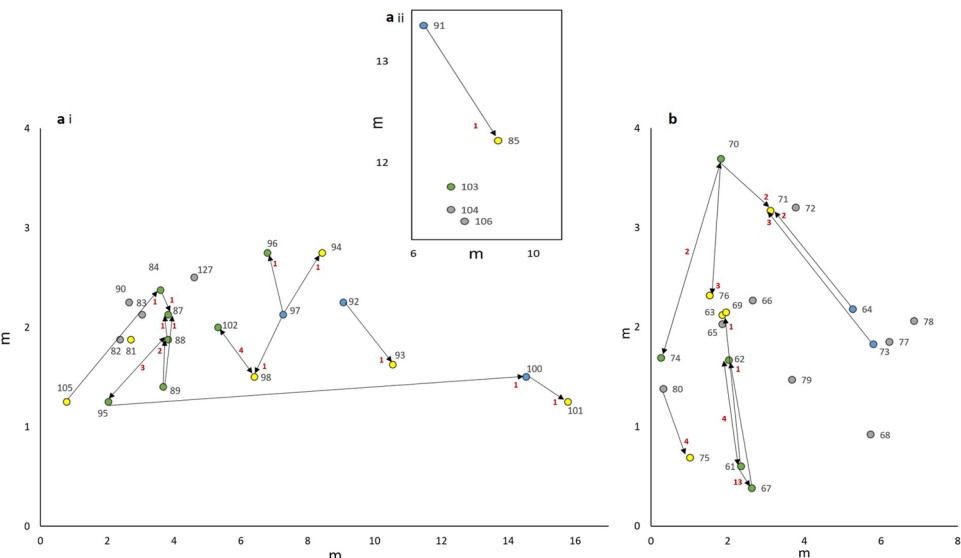

**Fig. 2 Direction of gyne transport in the ant *Cardiocondyla elegans*.** Location of colonies of the ant *Cardiocondyla elegans* in the collecting site RFRK (N 43° 55'43.9″, E 4°34'5.1″) in 2015 (**a** i and ii) and 2018 (**b**) in which the transport of female sexuals (gynes) by workers was observed. Shown are the exact localizations (in meters) of colonies used in the microsatellite analysis. Colonies marked by a yellow dot only received carried female sexuals (gynes) (yellow), colonies marked with blue only donated gynes, and colonies marked in green were both sources and recipients of gynes. Colonies not involved in gyne carrying are indicated in grey. Simple arrows indicate the direction of gyne transport. Double arrows indicate transport in both directions. Colored circles without arrows represent colonies for which the origin or destination of carried gynes could not be determined. The red numbers near the arrows indicate the total of transfers observed between individual nests. For the genetic analysis in 2015, we tried to maximize the number of nests involved, hence, the number of observed transfers between particular pairs of nests is lower than in 2018.

**Table 3 Genetic composition of colonies of the ant *Cardiocondyla elegans*.**

|  | Number of individuals, colonies | Relatedness (± SE) | Difference from 0.75 (single sample *t*-test) | Fixation coefficient (95% CI) | Difference from 0 (single sample *t*-test) |
|---|---|---|---|---|---|
| 2015 (five loci) | 247, 42 | 0.47 ± 0.04 | $t = 11.75\ p < 0.0001$ | 0.13 (0.11–0.15) | $t = 4.333\ p < 0.0001$ |
| 2018 (seven loci) | 111, 16 | 0.34 ± 0.05 | $t = 7.744\ p < 0.0001$ | 0.33 (0.14–0.60) | $t = 2.845\ p = 0.012$ |

Mean nestmate relatedness, difference from relatedness among full sisters (0.75) and fixation coefficient in populations of the ant *Cardiocondyla elegans* in different collecting years.

mating frequencies of 2.3–5.5. The fixation coefficient was significantly different from 0 and suggested that 37–66% of matings involved sibs (Table 3). Despite the often very close location of nest entrances, there was no evidence for neighboring nests belonging to the same colony or having been founded by fission or budding: genetic distances between colonies were not significantly correlated with spatial distances (Mantel test; $r = -0.012$, $p = 0.64$, Fig. 3).

The mean relatedness between carriers and their transported gynes was in the range of the relatedness among nestmates (2015: $0.39 \pm SE\ 0.08$, $n = 40$ pairs, *t*-test, $t = 1.27$, $p = 0.21$; 2018: $0.45 \pm SE\ 0.07$, $n = 27$ pairs, $t = 1.73$, $p = 0.09$), indicating that workers on average carry related gynes (Fig. 4). In 2015, we managed to take 40 complete samples consisting of carriers, gynes, and workers from source and recipient colonies. The mean relatedness of carriers and gynes to the source colony (based on six workers per colony) was significantly higher than 0 (carrier - source: $0.38 \pm SE\ 0.04$; $t = 9.5$, $p < 0.001$; gyne - source: $0.32 \pm SE\ 0.05$, $t = 6.4$, $p < 0.001$) and slightly lower than mean nestmate relatedness (carrier: $t = 2.25$, $p = 0.027$; gyne: $t = 3.32$, $p = 0.001$). The relatively high mean relatedness between carriers, queens, and the source colony indicates that workers typically carry sisters away from their joint natal colony (Fig. 4). In contrast, the mean relatedness of carriers to the recipient colony was around 0 (carrier: $-0.05 \pm SE\ 0.04$, $n = 40$) and not significantly different

from 0 ($t = 1.30$, $p = 0.20$). The mean relatedness of gynes to recipient colonies (only 2015; gyne: $-0.09 \pm SE\ 0.04$, $n = 40$) was lower than 0 ($t = 2.3$, $p = 0.03$). This indicates that gynes are preferably transported into colonies to which they are less related than to a random colony (Fig. 4).

Pairwise relatedness estimates based on a limited number of loci suffer from large standard errors. Indeed, the analysis of individual values of relatedness for carriers and gynes showed that there is considerable variation. Most gynes (26 of 40) were carried by a related carrier (relatedness ≥ 0.25, the value for halfsisters) from a related colony (relatedness ≥ 0.25, except in three cases, blue dots in the upper right part of Fig. 4) to an unrelated recipient colony (relatedness < 0.20, except in three cases, red dots in lower right part of Fig. 4). Other gynes appeared to be unrelated to the carrier (relatedness < 0.20, left part of Fig. 4) and/or the source colony (blue dots in lower left part of Fig. 4). Except for three carriers and six gynes, relatedness was higher to the source colony than to the recipient colony. In a pairwise comparison, the mean relatedness of a gyne to all colonies (mean 2, 1–4) passed during the transport ($-0.05 \pm SE\ 0.08$) was not significantly different from its relatedness to the recipient colony ($-0.14 \pm SE\ 0.10$; Wilcoxon matched pairs tests, $n = 9$, $T = 14$, $p = 0.31$).

Gynes and workers of *C. elegans* did not differ in heterozygosity at the studied microsatellite loci (27 pairs of carriers and

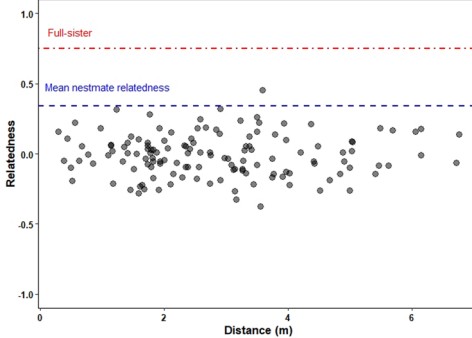

**Fig. 3 Association between genetic relatedness and spatial distance among nests in a population of the ant *Cardiocondyla elegans*.** Association between genetic relatedness and spatial distance among 16 colonies ($n = 7$ workers per colony) of the ant *Cardiocondyla elegans* in collecting site RFRK in 2018. The red dashed and dotted line represent the value of relatedness expected for full sisters (0.75), the blue dashed line represents the empirically determined mean relatedness among nestmate (2018: 0.34 ± 0.05, see Table 3).

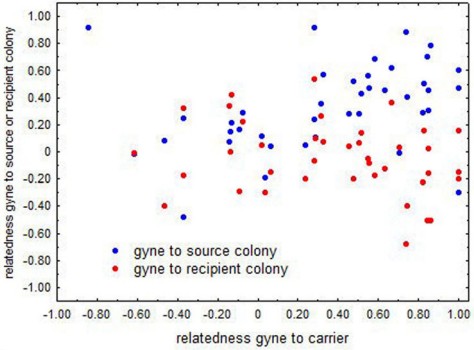

**Fig. 4 Genetic relationship between carried gynes, the carrier, and source and recipient colonies.** Association of the relatedness of gynes, which were carried by workers, to the source colony (blue) and to the recipient colony (red) with the relatedness between gyne and carrier ($n = 40$ pairs of gyne and carrier).

gynes from 2018, fixation coefficient: gynes 0.38 ± SE 0.12, workers 0.41 ± SE 0.14, $t = 0.23$, $p > 0.5$).

**Fate of carried gynes.** Observations in April 2016 showed that gynes had hibernated in the nests of established colonies. Excavation of 59 colonies revealed the presence of up to 130 winged or dealate gynes (mean = 25.5 ± SD 3.7), but no males. All 20 dissected gynes (nine winged, 11 dealate) from five different collecting sites (BN, CP, H, RFRK, and SM) were mated, i.e., they had sperm in their spermathecae, and 15 of them had begun to develop eggs in their ovaries. Solitary winged or dealate gynes were seen a few times dispersing on foot, but none was observed flying. Gyne carrying was observed only in a single instance.

Microsatellite genotyping showed that gynes collected in nests after hibernation from five colonies (site BN) were on average not related to the workers of the colony in which they were found (relatedness 0.06 ± 0.04; 2–19 gynes, total 62, and six workers per colony, two loci; not different from 0, $t = 1.5$; $p < 0.2$).

## Discussion

Animals have evolved a large number of strategies to avoid inbreeding, from sex-biased dispersal and delayed maturity to the avoidance of kin as mates[1,5]. Our study on the directed transport of female sexuals (gynes) by workers in the ant *Cardiocondyla elegans* adds a surprising new variant to these phenomena. Moreover, it raises the possibility of indirect or assisted female choice, i.e., workers might select the mating partner for their sexual sisters.

Our field observations revealed that the 2–3 mm long workers of *C. elegans* carry winged gynes in a straight line over distances of several meters and then dump them into the narrow entrance of another nest. According to microsatellite genotypes, the carriers mostly transport-related gynes from their natal nests to the nests of unrelated colonies. Dissections revealed that most of the carried gynes already contained sperm in their spermathecae. Together with the high inbreeding coefficient, this shows that they had already mated with related, non-dispersing wingless males in their natal nests. The transfer into another colony then allows them to outbreed with additional, alien males in "mating chambers" right below the nest entrance. The high effective paternity frequency estimated from worker genotypes (2.3–5.5 males) clearly supports multiple mating (see also[24]). Together with the occasionally observed low relatedness between carrier and gyne it also indicates that gynes may be repeatedly carried from one nest to another and probably mate with males from multiple colonies. After the mating period, gynes overwinter in an unrelated nest and disperse in spring, probably to found a new nest solitarily.

Assisted mating dispersal by workers represents an adaptive strategy to avoid inbreeding depression from recurrent sibmating[16]. In addition, the observation that workers transferred multiple gynes to a particular recipient colony while ignoring several closer nests might indicate that they select pools of potential mates for the carried gyne, thus exerting an intriguing type of indirect mate choice.

While the involvement of a third party in sexual selection is typical for zoophilic plants, which are pollinated by insects, birds, or bats[30], in animals it is known only from matchmaking and bridal exchange in human marriage markets, e.g., cultural exogamy and intermarriage[31]. Ant workers might in principle play a comparable selective role when mating occurs in or near the nest by granting only particular males access to the female sexuals. Such "reproductive interference" has been observed in Argentine ants (*Linepithema humile*), where workers attacked alien males entering their nests[32], and *Cataglyphis cursor*, where they more heavily attacked larger males close to the nest entrance[33,34]. However, at least in *Cataglyphis cursor*, worker aggression apparently did not affect the males' mating success, and there was also no evidence for mate choice by the gynes themselves[33]. Similarly, the queen-like morphology and rich glandular equipment of army ant males has been thought to be associated with worker control of mating[35], but behavioral observations[36], the very high mating frequency of queens[37], and the frequent hybridization in some species[38] do not support strong mate discrimination by workers.

Gyne carrying by workers in *C. elegans* might constitute an alternative way of indirect sexual selection, but our data do not yet unequivocally prove that workers actively choose particular colonies as recipients. For example, they might simply transfer multiple gynes into nests, which they previously happened to locate by chance while foraging. Our excavations of nests did not yet reveal differences between recipient and non-participating colonies, e.g., concerning the number of workers, males, or gynes. While gynes appeared to be significantly less closely related to recipient colonies than to random colonies, pairwise estimates of relatedness did not yet substantiate a difference between the relatedness of gynes to bypassed colonies and their relatedness to the recipient colony. Hence, additional experiments are necessary to determine if and how workers indeed choose specific colonies, how they find their way over the relatively large distances, and how they recognize the presence of suitable mates.

That colonies regularly accepted alien gynes is unexpected for a monogynous species (e.g.[39,40]). However, the observed trade of female sexuals appears to be advantageous for both source and recipient colonies in that it provides immobile male and female sexuals with opportunities to outbreed. Mating with close kin and subsequent mating with additional unrelated partners maintains considerable genetic heterogeneity in the offspring of the gyne. In contrast to ants with "social hybridogenesis," in which queens arise from mating between same-lineage partners or thelytokous parthenogenesis[41–45], queens of C. elegans do not depend on sperm from males of a different genetic lineage to obtain worker offspring. As we show, gynes and workers of C. elegans did not differ in heterozygosity at the studied microsatellite loci, suggesting that, unlike one taxon of the C. "kagutsuchi" complex[46], outbreeding is not associated with caste differentiation but instead counterbalances the genetic load from inbreeding. In addition, distributing gynes in a number of recipient colonies for hibernation might be a strategy of the source colony to spread the risk of losing all offspring in winter[24]. The river banks, in which Palearctic Cardiocondyla build their nests, maybe flooded in winter, and nest mortality is very high[47] (for mining bees and sweat bees nesting in similar places see[48,49]).

After overwintering, gynes leave the adoptive nest on foot, and several gynes, which had retained their wings after mating, were seen climbing up blades of grass and waving their wings, even though no flight activity was observed (unpublished observations). This shows that gynes are well capable of post-mating dispersal and raises the question about why gynes are carried by workers in summer rather than searching for males themselves. We suggest that this unique behavior reflects phylogenetic constraints resulting from the ancestral life history of Cardiocondyla, which combines polygyny, mating in the nest, post-copulatory dispersal on the wing or dependent colony founding by colony fission in tropical habitats, e.g.[15,19,23]. Derived monogyny and the loss of winged disperser males in temperate biomes[21] may have forced colonies to find novel ways to secure outbreeding opportunities. Rather than re-evolving winged disperser males, C. elegans appears to utilize the navigational knowledge of workers to promote gyne dispersal[50,51]. The carriers appear to "know" the exact position of suitable nests, probably from exploring the surroundings of their home nest by tandem running and foraging[18,20,52]. Furthermore, carrying nestmates is part of the workers' behavioral repertoire during nest moving[53]. In contrast, gynes of Palearctic Cardiocondyla are typically weak fliers, often with reduced wing muscles and wings[21,54] and do not have prior information about the location of alien colonies. Gyne-carrying might therefore more quickly and safely lead to a rendezvous with alien males than mate search by solitary gynes.

Our observations of assisted outbreeding and indirect mate choice in C. elegans raise the possibility that workers might be involved in sexual selection also in other social insects and highlight again that research at the intersection between social evolution and reproductive biology might reveal surprising novel facets of animal behavior.

## Methods

**Study species.** Cardiocondyla elegans[55,56] is a Mediterranean ant, which builds nests consisting of dozens of pea-sized cavities connected by narrow tunnels down to a depth of more than 1 m[24,54,57,58]. Colonies are monogynous and polyandrous, i.e., they contain a single, multiply-mated queen, and neither the queen nor its workers tolerate egg laying by additional queens[23]. Males are wingless, non-dispersing, and, in contrast to males of other Cardiocondyla species, mutually tolerant[23]. In Southern France, sexuals emerge between July and September.

Population structure and the transport of female sexuals (gynes) by workers were studied during 6 years (2014–2019), in seven sites in Languedoc-Roussillon (Southern France), between Beaucaire and Remoulins (BN: N 43° 50′ 38.1″, E 4° 36′ 59.5″; CP: N 43° 51′ 9.9″, E 4° 37′ 2.4″; FK: N 43° 55′ 39.8″, E 4° 34′ 18.1″; H: N 43° 55′ 2.7″, E 4° 35′ 4.2″; P: N 43° 56′ 31.0″, E 4° 33′ 34.5″; RFRK: N 43° 55′ 43.9″, E 4° 34′ 5.1″; SM: N 43° 51′ 10.5″, E 4° 37′ 2.2″). All sites are sparsely vegetated, sandy areas on the banks of rivers Gardon and Rhône, except for site "P", which is an unpaved sandy parking lot near the city center of Remoulins.

**Field observations.** Nests were located by following foragers back to the 1 mm wide nest entrance, marked with a colored flag with a number, and observed subsequently. Source colonies were identified by observing a worker carrying a gyne leaving a nest. Carrying workers were followed until the gyne was to be introduced into the entrance of the recipient colony. We observed in total 453 pairs of carriers and carried gynes, of which 357 pairs could be collected with an aspirator. For 86 pairs, and 55 additional pairs, which were not collected, we could follow the whole transport from the source colony to the entrance of the recipient colony. Of the total pairs collected, 113 pairs were stored in 100% ethanol, the remaining gynes were kept alive in laboratory nests for other studies or the creation of laboratory colonies. In addition, 20 workers from each colony located in the population were captured to determine the genetic relatedness of carrying pairs to source and recipient colonies and, in a few cases, also to colonies passed during their carrying trips. Several gynes were killed by freezing and later dissected with the help of fine forceps under a binocular microscope to determine their reproductive status. Overall, the ants were collected with the approval of the Access and Benefit-Sharing Clearing-House (ABSCH). We obtained a certificate of compliance allowing the collection of Cardiocondyla elegans in Gard (France)[59].

**Microsatellite analyses.** DNA was extracted from whole bodies with CTAB method (modified from[60]) and diluted in 25 µl of TE buffer. Samples from 2015 were genotyped at 1–5 microsatellite loci (CE2-3A, CE2-4A, CE2-4E, CE2-5D, CE2-12D[61]), samples from 2018 at up to seven loci, including Card 8 and Cobs 13[13,62] (see supplementary table 1). For several analyses, one locus (CE2-5D) was excluded because almost all individuals were homozygous at the same allele. Due to this and amplification failures, genotypes were missing at one or several loci in the samples from 2015 (missing genotypes per worker from a colony, median 1, quartiles 1, 1; missing genotypes per carrier and gyne, median 0, quartiles 0, 1) while the genotypes from 2018 were almost fully complete (median, quartiles 0, 0, 0 missing genotypes). PCRs were performed in a 20 µl reaction volume using 1 µl of DNA with 19 µl of master-mix (7 µl $H_2O$, 10 µl GoTaq, 1 µl of each forward and reverse primers). Samples were amplified following Lenoir et al. (2005) with an initial denaturation step at 94 °C for 3 min, 40 cycles at 94 °C for 45 s, with an annealing temperature according to the primer for 45 s (Supplementary Table 1), followed by a step of 72 °C for 45 s, and a final extension step at 72 °C for 7 min. Microsatellite DNA was analyzed using an ABI Prism genetic analyzer with 0.1–0.15 µl of PCR product, 25.15 µl of ABI master-mix (25 µl formamide, 0.15 µl standard size T486). Allele size was determined using GeneScan® 500 TAMRA dye size standard and GeneScan® 3.1 software (Applied Biosystems).

**Statistics and reproducibility.** Genetic data were analyzed using the software package GDA[63]. Relatedness was estimated following[64] using the software Relatedness v4.2[65] and GenAlEx v6.51b2[66] with standard errors of means obtained by jackknifing by groups. Confidence limits for the fixation coefficients were obtained by bootstrapping over loci (5000x). From fixation coefficients F, we calculated the proportion of sib-mating α using F = α/(4–3 α)[67,68]. The Mantel test was made using R software-4.0.3. Other statistical tests were conducted using Statistica v6.0 (StatSoft, Tulsa, OK).

In 2015, colonies, gynes, and carriers were collected in four sites (BN, CP, RFRK, SM) with an $F_{ST}$ value of 0.042 ± 0.051 (jackknifed over sites). This value was not significantly different from 0 (single sample t-test, $t = 0.823$, $p > 0.1$), suggesting that the four sites belong to a single population extending along the rivers. In 2018 we concentrated on a single collecting site, RFRK.

**Reporting summary.** Further information on research design is available in the Nature Research Reporting Summary linked to this article.

## Data availability

The genetic data, data on mapped and excavated colonies and pairs of gynes and carriers observed are available in Figshare[26]. All other data are available upon request to the corresponding author; Mathilde Vidal at mathilde.vidal@biologie.uni-regensburg.de.

## Code availability

The R code for the Mantel test is available in Figshare[26].

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

## Acknowledgements
This work was supported by DAAD (PROCOPE 57128510) and DFG (He1623/42). We thank Jean-Luc Mercier, Larissa Kalb, Sarah Koller, Christopher Schreier, Çiğdem Ün and Tobias Wallner for help in the field as well as Sarah Koller and Anaïs Chanson for taking part in the analysis of microsatellites.

## Author contributions
J.G., F.K., A.S., and J.H. carried out the experiment in 2014, 2015, and 2016. M.V. conducted fieldwork and sequenced microsatellites in 2017 and 2018, analyzed the data and together with C.L. and J.H. wrote the manuscript.

## Funding

## Competing interests
The authors declare no competing interests.
