## [Peer Review File · Communications Biology]

Reviewers' comments:

Reviewer #1 (Remarks to the Author):

This manuscript documents a remarkable behavior in the inbred ant *Cardiocondyla elegans*. Queens of this species typically mate with their nestmate brothers. The authors combined observations in the field and genetic analyses to show that workers carry their reproductive sisters in foreign nests, where they may encounter unrelated resident males. This mechanism may allow queens to occasionally outbreed and ultimately prevent inbreeding depression. This is a fascinating and overall well-executed study.

My only concern is that the sampling strategy and sizes are often obscure. For example:

- Were all sites or only the RFKR population used for genetic analyses?
- Were relatedness values estimated for each site/year separately?
- What are the sample sizes for analyses I204-213?
- Is there any data for the populations FK and P mentioned in the methods?
- Were all the 290 collected pairs (I113) used for genetics? Relatedness seems to have only been estimated for 67 of these (I202-203).
- Some sites were visited several times. Do colonies remain at the same place from one year to another?
- I do not understand the statement I506-508. The total sums (122 and 406) seem to match the figures found in the main text.
- I123: How many individuals were genotyped at just 2 loci? A percentage of missing data may be more informative.

Minor comments:

- I64-67: CSD is widespread in ants, but sl-CSD has not been demonstrated in any species so far. I would rephrase the sentence.
- I161: Table 2 -> Table1
- I187: Is male carrying rare or just difficult to observe? Could it take place earlier in the season? If male carrying regularly occurs, an alternative scenario to the one proposed could be that queens mate with both natal and alien males in their natal nest and are then transported in a recipient nest (where they may not mate again).
- I204: The sentence is incomplete.
- I227-228: This sentence is a bit ambiguous. I assume you mean they have developed eggs in their ovaries?
- I289-291: It seems like this analysis could have done with the present data (?)
- Figure3: The mean nestmate relatedness (0.34) is perhaps more interesting to represent than the relatedness between full sisters.

Reviewer #2 (Remarks to the Author):

*** Copied in by editor ***

Communication Biology 2020 – COMMSBIO-20-3445-T.

Royal matchmaking: ant workers carry their sexual sisters into alien nests to promote outbreeding

M. Vidal, F. Königseder, J. Giehr, A. Schrempf, C. Lucas, J. Heinze

In Hymenoptera, inbreeding may be particularly detrimental because it typically results in production of sterile diploid males. In this paper, the authors nicely show that in the highly inbred ant species *Cardiocondyla elegans*, workers may promote outbreeding by carrying their reproductive daughters to unrelated nests where they can mate with wingless males. Such a “third party sexual selection” is fascinating – even though probably a rare phenomenon in ants

and other social Hymenoptera since in most species sexuals take part in large mating flights. This submission is a follow-up of a previous paper (Lenoir et al. 2007 - *Molecular Ecology* 16: 345-354), where the authors showed that "workers of *Cardiocondyla elegans* carry female sexuals from their natal nest over several meters and drop them in the nest of another, unrelated colony to promote outbreeding". Here, the authors combined behavioral observations and genetic analyses that support the role of workers in outbreeding; furthermore, they present additional results on the transport of winged queens and relatedness among the parties involved. The study is sound, sampling is good and the results are interesting.

I have however some concerns with the manuscript.

1. The fundamental idea of the paper is that the transport of alate queens to alien nests promotes outbreeding. The results presented are converge towards such a conclusion. However, I regret that a detailed comparison of the content of the spermatheca between gynes during transport and after transport/hibernation is lacking. Such comparison would allow to estimate the relatedness between the alate queens and the sperm stored in their spermatheca, and to test directly for an increase (i) in queen mating frequency and (ii) in genetic diversity of sperm cells. Males of Hymenoptera being haploid, this is straightforward. Alternatively, comparison of parent-offspring genetic combinations from mated gynes before transport and after hibernation could also do the job. Such analyses would show that gynes effectively remate with foreign males after their transfer to unrelated colonies. They would add a nice complement to the paper of Lenoir et al. (2007) and help establish unequivocally the role of the third party in sexual selection.

2. The manuscript contains several inaccuracies and should be improved.

- lines 77-81 – How colony queen number affects colony kin structure and level of inbreeding is unclear.

- lines 204-209 – This section is weird. It is written that "The relatedness of carriers and gynes to the source colony was slightly (though significantly) lower than mean nestmate relatedness, meaning that workers typically carry sisters away from their joint natal colony".

First, how do the authors account for the difference between this result and those reported in the previous paragraph showing that "the relatedness between carriers and their transported gynes was not different from the relatedness among nestmates"?

Second, how these relatedness values allow the authors to conclude that "workers typically carry sisters away from their joint natal colony" is unclear.

- lines 211-213 – "gynes are preferably transported into colonies to which they are less related than to a random colony". The data do not allow such inference. The authors must give sample size and range of relatedness values; in addition, I strongly suggest them to perform permutation tests to confirm (or not) outbreeding.

- Abstract and lines 259-261 – Comparison with human matchmaking is meaningless.

Matchmaking in humans belongs primarily to cultural and/or political affairs, rather than on population genetics. Using such a comparison gives the feeling that the authors oversell their work.

Minor comments

l.64 – Not only social - but all Hymenoptera

l. 65 – Indicate this is sl-CSD (not defined l. 72)

l. 67 – Ref.10 is irrelevant – it does not concern social Hymenoptera

l. 67-69 – Inbreeding avoidance in social insects is typically performed thanks to large nuptial flights that greatly reduce the probability of sib-mating. There are also several papers on inbreeding avoidance based on kin recognition.

l. 145 – The authors mapped 61 colonies on RFRK between 2014 and 2019. How do they make sure that they have not sampled the same colonies twice (or more) in case they moved?

l. 147 – m² (meter squared)

l. 161 – Table 2, I guess

l. 189-190 – Were the prepupae and pupae carried sexual or worker pupae?

Figure 2 – Please, increase scale to make the figure clearer for readers; eventually, add an inset for nests 85,91,103,104,106.

Reviewers' comments:

Reviewer #1 (Remarks to the Author):

This manuscript documents a remarkable behavior in the inbred ant *Cardiocondyla elegans*. Queens of this species typically mate with their nestmate brothers. The authors combined observations in the field and genetic analyses to show that workers carry their reproductive sisters in foreign nests, where they may encounter unrelated resident males. This mechanism may allow queens to occasionally outbreed and ultimately prevent inbreeding depression. This is a fascinating and overall well-executed study.

My only concern is that the sampling strategy and sizes are often obscure. For example:

1.1_ Were all sites or only the RFRK population used for genetic analyses?

The main results are from 2015 and 2018. In 2015, we used samples from four collecting sites (RFRK, BN, CP, and SM) for observations and the analysis of nestmate relatedness and the relatedness between gynes, carriers, source and recipient colonies. Across the samples from these sites, F_{st} was 0.042 ± 0.0511 , i.e., overlapping 0. This means that we could pool samples from different collecting sites in the analysis. This is now indicated in the text.

The 2014 data came from six populations, the above plus BO and P. The analysis gave an F_{st} of 0.1, which was different from 0. This means that collecting sites should have been analysed separately, which, however, was not possible because of small sample sizes per site. As the 2014 value for relatedness was only listed in table 3 and is not discussed in more detail we decided to exclude this value from the manuscript.

In 2018 we collected samples only in one site. This again is now indicated in the text.

1.2_ Were relatedness values estimated for each site/year separately?

Samples from different years were analysed separately (as already indicated in the text, e.g., line 206 and Table 3).

1.3_ What are the sample sizes for analyses I204-213?

The sample size, $n = 40$, is now given.

1.4_ Is there any data for the populations FK and P mentioned in the methods?

We now added these sampling sites to the table. While gyne carrying was observed in these sites and colonies were excavated for other studies, samples from these populations were not used for the genetic analyses and had therefore not been listed in the previous version of the manuscript.

1.5_ Were all the 290 collected pairs (I113) used for genetics? Relatedness seems to have only been estimated for 67 of these (I202-203).

Only 113 pairs were stored in EtOH after collecting, the others were used to set up colonies for future analyses of the interrelation between mating frequency and productivity. Of these 113 pairs, we indeed used 40 for which source and recipient colonies were known (2015) and a random sample of 27 carrier and carried gyne from 2018 to estimate relatedness.

1.6_ Some sites were visited several times. Do colonies remain at the same place from one year to another?

This ant species preferentially nests in river banks, in sandy soil, which during winter may be flooded. During his doctoral thesis, J.C. Lenoir found that "40% of the nests died during the flood event. The death of the nests was not directly due to the flood destroying the sandy deposits but more probably because of flooding of the badly constructed chambers and/or ant asphyxiation. [...] New nests are

founded by winged females who disperse from the surviving nests. These new nests appear regularly spaced from the established ones (already in activity or still cryptic at the founding event)."

Lenoir, J.-C. Structure sociale et stratégie de reproduction chez Cardiocondyla elegans. (Université François Rabelais - Tours, 2006).

Hence, the population structure and the location of individual colonies will change considerably from year to year. In addition, the collecting sites covered relatively large areas and we never collected at exactly the same spot two years in a row. Finally, at the end of each field collection, most observed colonies were excavated.

This all makes it highly unlikely that the same colony was investigated repeatedly in different years.

1.7_ I do not understand the statement l506-508. The total sums (122 and 406) seem to match the figures found in the main text.

We corrected this mistake and now include samples from all populations in the table.

1.8_ l123: How many individuals were genotyped at just 2 loci? A percentage of missing data may be more informative.

Genotype data are mostly missing for the samples from 2015. This is now indicated in the text as follows "...genotypes were missing at one or several loci in the samples from 2015 (missing genotypes per worker from a colony, median 1, quartiles 1, 1; missing genotypes per carrier and gyne, median 0, quartiles 0, 1) while the genotypes from 2018 were almost fully complete (median, quartiles 0, 0, 0 missing genotypes).."

Minor comments:

1.9_ l64-67: CSD is widespread in ants, but sl-CSD has not been demonstrated in any species so far. I would rephrase the sentence.

Changed to "In ants and many other Hymenoptera inbreeding is particularly detrimental^{6,7}. Due to their standard mechanism of sex determination, 50% of a female's eggs that are fertilized by sperm from a brother may develop into non-viable or sterile diploid males instead of diploid females".

1.10_ l161: Table 2 -> Table1

Corrected as suggested.

1.11_ l187: Is male carrying rare or just difficult to observe? Could it take place earlier in the season? If male carrying regularly occurs, an alternative scenario to the one proposed could be that queens mate with both natal and alien males in their natal nest and are then transported in a recipient nest (where they may not mate again).

Carrying of males would have been detected easily, due to their particular colour (orange). The fact that we did not observe it more often clearly shows that it is a rare event.

1.12_ l204: The sentence is incomplete.

Corrected as suggested. The sentence reads now: "In 2015, we managed to take 40 complete samples consisting of carriers, gynes, and workers from source and recipient colonies"

1.13_ l227-228: This sentence is a bit ambiguous. I assume you mean they have developed eggs in their ovaries?

Changed as suggested.

1.14_ -l289-291: It seems like this analysis could have been done with the present data (?)

This analysis was indeed based on data from the present study and the sentence has therefore been transferred to the results.

1.15_Figure3: The mean nestmate relatedness (0.34) is perhaps more interesting to represent than the relatedness between full sisters.

We added a line showing the mean relatedness in the colonies of C. elegans.

Figure 3:

Reviewer #2 (Remarks to the Author):

In Hymenoptera, inbreeding may be particularly detrimental because it typically results in production of sterile diploid males. In this paper, the authors nicely show that in the highly inbred ant species *Cardiocondyla elegans*, workers may promote outbreeding by carrying their reproductive daughters to unrelated nests where they can mate with wingless males. Such a “third party sexual selection” is fascinating – even though probably a rare phenomenon in ants and other social Hymenoptera since in most species sexuals take part in large mating flights.

This submission is a follow-up of a previous paper (Lenoir et al. 2007 - *Molecular Ecology* 16: 345-354), where the authors showed that “workers of *Cardiocondyla elegans* carry female sexuals from their natal nest over several meters and drop them in the nest of another, unrelated colony to promote outbreeding”. Here, the authors combined behavioral observations and genetic analyses that support the role of workers in outbreeding; furthermore, they present additional results on the transport of winged queens and relatedness among the parties involved. The study is sound, sampling is good and the results are interesting.

I have however some concerns with the manuscript.

2.1.1_ The fundamental idea of the paper is that the transport of alate queens to alien nests promotes outbreeding. The results presented converge towards such a conclusion.

However, I regret that a detailed comparison of the content of the spermatheca between gynes during transport and after transport/hibernation is lacking. Such comparison would allow to estimate the relatedness between the alate queens and the sperm stored in their spermatheca, and to test directly for an increase (i) in queen mating frequency and (ii) in genetic diversity of sperm cells. Males of Hymenoptera being haploid, this is straightforward. Alternatively, comparison of parent-offspring genetic combinations from mated gynes before transport and after hibernation could also do the job. Such analyses would show that gynes effectively remate with foreign males after their transfer to unrelated colonies. They would add a nice complement to the paper of Lenoir et al. (2007) and help establish unequivocally the role of the third party in sexual selection.

In an ideal world we could have done this. Given that gynes are presumably transported multiply among colonies, the difference in spermatheca content between transported and hibernated gynes would probably not have been large. Alternatively, we could have checked spermatheca content early and late in the season, however, this would have required a much longer stay in the field.

We know from the inbreeding coefficient that gynes mate with their brothers and the genotypes of offspring clearly support this. At the same time, the genotypes of workers in natural colonies support multiple mating also with unrelated males. Hence, an analysis of spermathecal content would not have added information.

The manuscript contains several inaccuracies and should be improved.

2.1.2_ lines 77-81 – How colony queen number affects colony kin structure and level of inbreeding is unclear.

We re-wrote the sentences to make it clearer that outbreeding in polygynous species can be achieved by adoption of alien queens while this is not the case in monogynous species.

2.2_ lines 204-209 – This section is weird. It is written that “The relatedness of carriers and gynes to the source colony was ... slightly (though significantly) lower than mean nestmate relatedness ..., meaning that workers typically carry sisters away from their joint natal colony”.

2.2.1*First, how do the authors account for the difference between this result and those reported in the previous paragraph showing that “the relatedness between carriers and their transported gynes was not different from the relatedness among nestmates”?

We originally tried to explain this in the paragraph about the large variation of pairwise relatedness estimates. Relatedness values for pairs of gynes and carriers were more variable (SE 0.08) than both nestmate relatedness (SE 0.04) and the relatedness of gynes or carriers to the source colonies (SE 0.04 and 0.05, respectively). Hence, though the average relatedness of gynes and carriers is in the range of nestmate relatedness, there are a number of pairs in which gynes and carriers are unrelated. This leads to the apparent contradiction between the two estimates.

2.2.2*Second, how these relatedness values allow the authors to conclude that “workers typically carry sisters away from their joint natal colony” is unclear.

We know where transport started and where it ended. Together with the relatedness estimate we can safely conclude that in most cases workers carry gynes away from their natal colony. This is now explained in more detail in the text.

2.3_ lines 211-213 – “gynes are preferably transported into colonies to which they are less related than to a random colony”. The data do not allow such inference. The authors must give sample size and range of relatedness values; in addition, I strongly suggest them to perform permutation tests to confirm (or not) outbreeding.

The range of pairwise relatedness estimates can be seen in the new Figure 4, which in contrast to the original figure now presents values of gyne-colony relatedness. The red dots in this figure show that most gynes are transported to colonies to which they are related by a value of < 0.2. This is also discussed in the paragraph starting in line 224.

Standard errors from estimates of F-values were obtained by bootstrapping with 5000 permutations. Testing a normally distributed sample of means resulting from this procedure against an expected value, e.g., 0 in this case, is the standard statistics to determine whether the difference is significant or not.

2.4_ Abstract and lines 259-261 – Comparison with human matchmaking is meaningless. Matchmaking in humans belongs primarily to cultural and/or political affairs, rather than on population genetics. Using such a comparison gives the feeling that the authors oversell their work.

*We agree that matchmaking in humans is based on culture, while gyne carrying in *C. elegans* is an evolved trait. Nevertheless, the two phenomena resemble each other, regardless of the underlying proximate and ultimate causes, while there are no comparable cases reported from animals. We therefore would like to keep the respective sentences.*

Minor comments:

2.5_ l.64 – Not only social - but all Hymenoptera

In parasitoid wasps, fig wasps and others kin regularly mate without the production of diploid males. Nevertheless, we now write “In ants and many other Hymenoptera”

2.6_ l. 65 – Indicate this is sl-CSD (not defined l. 72)

As mentioned by reviewer 1, the exact mode of CSD in ants is not known. We therefore removed this term from the manuscript.

2.7_l. 67 – Ref.10 is irrelevant – it does not concern social Hymenoptera

We replaced this reference by Doums et al. 2013, Behav Ecol Sociobiol 67, 1983–1993

2.8_l. 67-69 – Inbreeding avoidance in social insects is typically performed thanks to large nuptial flights that greatly reduce the probability of sib-mating. There are also several papers on inbreeding avoidance based on kin recognition.

We added nuptial flights and cited Boomsma et al. 2005 for more detail about mating behaviour in social insects.

Several publications show that there is no inbreeding avoidance due to kin recognition; hence, we did not take up this suggestion. See also Oppelt et al. (2008). The significance of intercolonial variation of cuticular hydrocarbons for inbreeding avoidance in ant sexuals. Animal Behaviour, 76(3), 1029–1034. <https://doi.org/10.1016/j.anbehav.2008.05.020>

2.9_l. 145 – The authors mapped 61 colonies on RFRK between 2014 and 2019. How do they make sure that they have not sampled the same colonies twice (or more) in case they moved?

See answer to the comment by reviewer 1 concerning the same topic.

2.10_l. 147 – m² (meter squared)

Changed as suggested.

2.11_l. 161 – Table 2, I guess

Corrected as suggested.

2.12_l. 189-190 – Were the prepupae and pupae carried sexual or worker pupae?

Unfortunately, we did not collect the samples and it is difficult to determine the sex and caste of pupae from a distance. In any case, the transport of pupae and prepupae was observed only in a few instances.

2.13_Figure 2 – Please, increase scale to make the figure clearer for readers; eventually, add an inset for nests 85,91,103,104,106.

Changed as suggested.

Figure 2:

REVIEWERS' COMMENTS:

Reviewer #2 (Remarks to the Author):

This is my second review of the manuscript as Reviewer #2.

The authors have taken into account - and satisfactorily modified their manuscript according to - most of the reviewers comments. Thus, I have no objection to this revised manuscript being accepted for publication.

However, let me stress that I am not satisfied with the response to comment 1 (2.1.1 according to the authors' numbering). According to the standards of Communication Biology, I expected a direct demonstration of the "third party sexual selection", rather than inferences based on workers' genotypes. Invoking "an ideal world" is irrelevant, here.